# Bidirectional alterations in brain temperature profoundly modulate spatiotemporal neurovascular responses in-vivo

Luke W. Boorman[1,6], Samuel S. Harris[2,6], Osman Shabir [1,3,4,5], Llywelyn Lee[1,4,5], Beth Eyre[1,4,5], Clare Howarth [1,4,5] & Jason Berwick[1,4,5 ✉]

Neurovascular coupling (NVC) is a mechanism that, amongst other known and latent critical functions, ensures activated brain regions are adequately supplied with oxygen and glucose. This biological phenomenon underpins non-invasive perfusion-related neuroimaging techniques and recent reports have implicated NVC impairment in several neurodegenerative disorders. Yet, much remains unknown regarding NVC in health and disease, and only recently has there been burgeoning recognition of a close interplay with brain thermodynamics. Accordingly, we developed a novel multi-modal approach to systematically modulate cortical temperature and interrogate the spatiotemporal dynamics of sensory-evoked NVC. We show that changes in cortical temperature profoundly and intricately modulate NVC, with low temperatures associated with diminished oxygen delivery, and high temperatures inducing a distinct vascular oscillation. These observations provide novel insights into the relationship between NVC and brain thermodynamics, with important implications for brain-temperature related therapies, functional biomarkers of elevated brain temperature, and in-vivo methods to study neurovascular coupling.

[1] Department of Psychology, University of Sheffield, Sheffield, UK. [2] UK Dementia Research Institute at University College London, University College London, London, UK. [3] Department of Infection Immunity and Cardiovascular Disease, University of Sheffield, Sheffield, UK. [4] Neuroscience Institute, University of Sheffield, Sheffield, UK. [5] Healthy Lifespan Institute, University of Sheffield, Sheffield, UK. [6]These authors contributed equally: Luke W. Boorman, Samuel S. Harris. ✉email: j.berwick@sheffield.ac.uk

Neurovascular coupling is a vital homoeostatic mechanism that subserves numerous critical functions in the healthy brain, including delivery of oxygen and glucose to activated regions, clearance of waste materials and metabolic by-products, neuroimmune trafficking, and regulation of cerebral temperature[1]. Preserved neurovascular coupling is a fundamental assumption underpinning the inference of neuronal activation from perfusion-related neuroimaging signals, such as blood oxygen level dependent (BOLD) functional magnetic resonance imaging (fMRI)[2]. Impaired neurovascular coupling, in turn, has engendered particular interest of late, with recent reports implicating deficits as playing a key role in the progression, and perhaps initiation, of neurodegenerative disorders such as Alzheimer's disease[3–5], and underscoring the potential of neurovascular unit deficits as novel targets for therapy and sensitive biomarkers of early disease.

Brain temperature is regulated by a complex interplay between cerebral metabolism, blood flow, and core body temperature, such that in healthy human subjects, heat production in activated brain regions due to increased metabolic rate is dissipated by the inflow of core-temperature blood during functional hyperaemia[6,7]. Pathological alterations in brain temperature, on the other hand, have become increasingly recognised as an important feature in several disorders including neurodegenerative disease, epilepsy, brain injury and stroke [8,9]. Age-dependent decline in cerebral metabolism is associated with a concomitant reduction in brain temperature[10], and diminished brain temperature in Parkinson's disease patients has been ascribed to impaired mitochondrial biogenesis[11,12], with subjects with mitochondrial disease exhibiting cerebral hypothermia as a result of defective oxidative phosphorylation[13]. Fever (pyrexia) following stroke is also associated with increases in morbidity and mortality[14], and is frequently observed following traumatic brain injury[15] and linked to increased neurological severity and length of stay in intensive care units[16]. Increases in brain temperature are concomitantly observed during seizure activity[17,18] and fever-induced seizures are the most prevalent pathological brain activity during development, with a disproportionate number of adult patients with medial temporal lobe epilepsy having had febrile seizures during childhood[19,20]. These reports, and others, have led to considerable recent interest in manipulating brain temperature as a therapeutic strategy to improve neurological disease outcomes, although clinical studies have reported mixed successes potentially as a result of a lack of consensus on optimal interventional protocols[21–24]. While there is substantial evidence that changes in brain temperature alter vascular-associated responses, such as the affinity of haemoglobin to oxygen (and thus blood oxygen saturation)[7], blood–brain barrier permeability[25], and cerebral blood flow, as well as neurometabolic rates[18], very little is known on the influence of brain temperature on the spatiotemporal evolution of neurovascular coupling. Addressing this important research gap is critical to (1) elucidating how pathological alterations in brain temperature exacerbate adverse clinical outcomes in a variety of brain disorders, (2) developing rational and effective therapeutic approaches based on brain temperature modulation, and (3) enabling a more precise interpretation of BOLD fMRI related signals, in terms of underlying neuronal activation, in health and disease.

Accordingly, here we sought to systematically interrogate how bidirectional (i.e., hypo- and hyper-thermic) modulation of brain temperature impacts neurovascular coupling and test the hypothesis that such alterations would dynamically alter the relationship between evoked neural activity and haemodynamics, in particular the temporal evolution of both variables and the dynamics of functional hyperaemia and washout. To this end, we deployed concurrent multi-modal measurements of laminar neural activity, oxygenated and deoxygenated haemoglobin concentration, tissue oxygenation and temperature,

and graded modulation of local brain temperature. Using the well-characterised rat somatosensory cortex as our model system, we reveal that progressive brain cooling results in an increased delay in the onset of neurovascular coupling and increased prominence of a robust transient increase in deoxyhaemoglobin ('deoxy-dip') during precisely defined whisker stimulation. Conversely, we found that elevated cortical temperatures were associated with the emergence of a pathological low-frequency oscillation. Together with the finding of an inverted U-shaped relationship between modulated temperature and magnitude of evoked neuronal activity and hemodynamics (and coupling), these data thus provide important insights into the association between neurovascular coupling and brain thermodynamics, and have important implications for our understanding of the role of brain temperature in pathogenic processes following events such as stroke, brain injury and seizures, and its potential value as a target for therapy and diagnosis[26].

## Results

**Multi-modal approach for precise manipulation of cortical temperature and measurement of neurovascular function.** We employed a novel method to modulate cortical temperature with fine control (Fig. 1a, see Methods) alongside multi-modal measurement of cortical hemodynamics, laminar neural activity, temperature, and tissue oxygenation (Fig. 1b, see Methods). Our temperature modulation approach using a skull-attached chamber and coil yielded stable, and linearly related, changes in baseline cortical temperature (Fig. 1c), and induced expected variations in baseline tissue oxygenation (pO$_2$), in the form of non-linear monotonically increasing relationship (Fig. 1d and see also Supplementary Tables 1, 2). The observed decrease in pO$_2$ with decreasing brain temperature emerges despite blood oxygen saturation increasing concomitantly (as demonstrated by baseline spectroscopy calculations detailed in Methods), since lowering brain temperature increases the affinity of haemoglobin to oxygen[27]. In addition, our methodology was capable of discerning small but statistically significant changes in evoked cortical temperature to 2 s and 16 s whisker stimulation during modulation of cortical temperature (single factor ANOVA, 2 s, results $F = 9.93$, $p = 2.9 \times 10^{-6}$, 16 s, $F = 33.45$, $p = 6.7 \times 10^{-9}$). Of note, an inverted U-shaped relationship (of the form $y = a + bx + cx^2$, curve fitting implemented using non-linear least squares, goodness-of-fit 2 s, 0.9; 16 s, 0.86) was observed between baseline and stimulus-evoked changes in cortical temperature (Fig. 1e). This association indicated that influx of core-temperature blood during functional hyperaemia produced a cooling effect in cortex when the cortical temperature exceeded that of the core (maintained at ~37 °C using a homoeothermic blanket), as denoted by a $y = 0$ crossing at 37.6 ± 0.19 °C (Fig. 1e, pooled stimulation conditions), and an opposing warming effect when the cortical temperature was reduced relative to the core. These results demonstrate the effective manipulation of brain temperature using our methodology which enabled the subsequent investigation into the association between brain temperature and sensory-evoked neural and vascular responses.

**Non-linear relationship between cortical temperature and neural activity during sensory processing.** Evoked local field potential (LFP) responses to 2 and 16 s whisker stimulation during cortical temperature manipulation were averaged to produce a mean impulse response across the 1500 μm depth spanned by the multi-channel electrode (Fig. 2a). Following extraction of stimulation LFP timeseries from granular layers (400–900 μm), the systematic modulation of cortical temperature was found to significantly affect the amplitude of the evoked LFP negative deflection (single factor

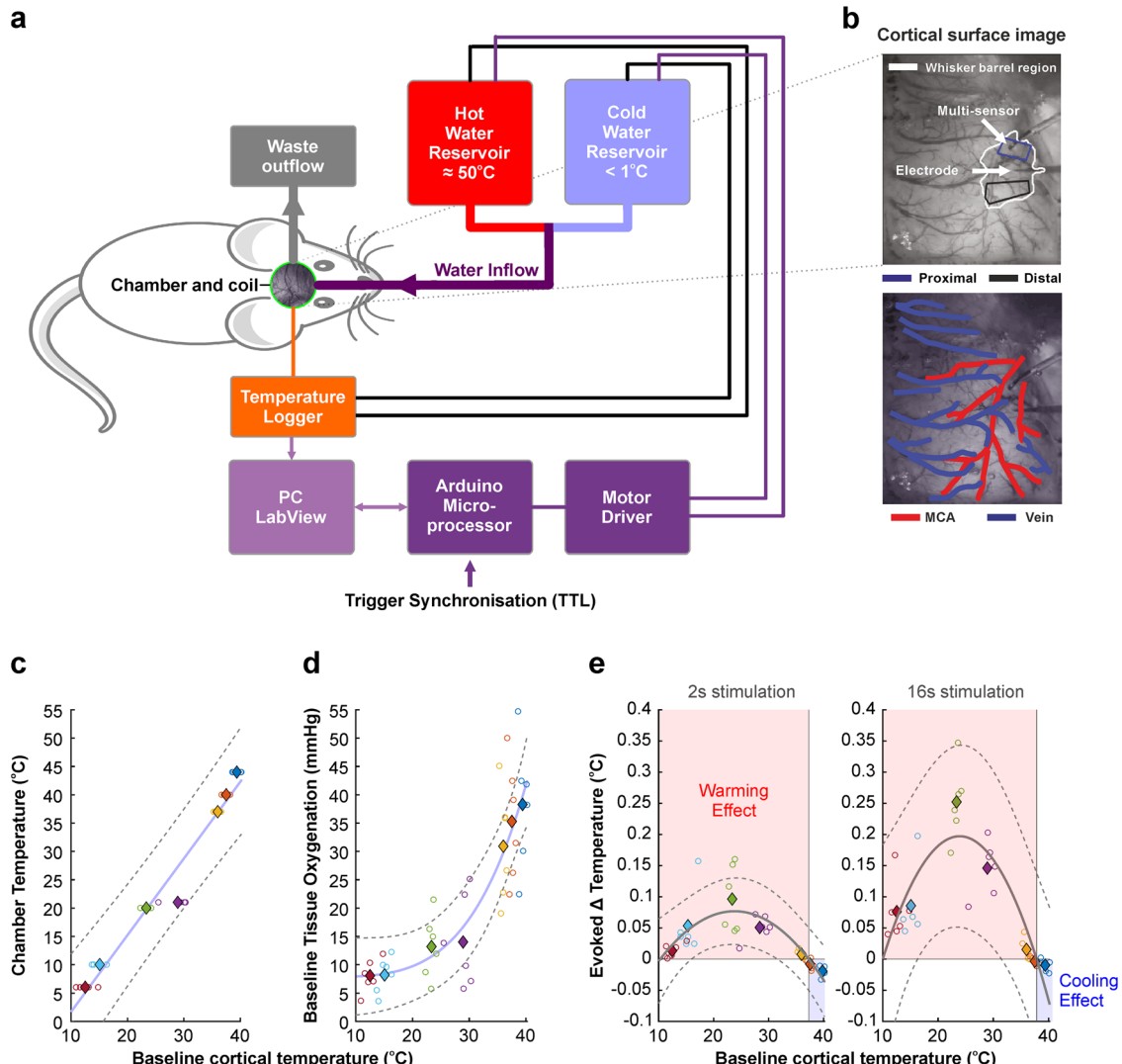

**Fig. 1 A novel methodology to precisely modulate cortical temperature and monitor neurovascular responses in the rat. a** Schematic of cortical cooling approach using computer-based proportional integration and differentiation (PID) to thermally manipulate cortical regions. **b** Digital images of cortical surface illustrating the positioning of the multi-channel electrode and multi-sensor tissue oxygenation and temperature probe in whisker barrel cortex (top), and identification of major surface arteries and veins (bottom). **c** Manipulation of cortical temperature using a skull-attached fluidic chamber induced reliable and stable alterations in cortical temperature. **d** Changes in cortical temperature canonically altered baseline tissue oxygenation due to changes in the affinity of haemoglobin to oxygen. **e** 2 s and 16 s whisker stimulation-induced alterations in cortical temperature that varied as a function of baseline cortical temperature, such that influx of blood with functional hyperaemia induced a cooling effect when the cortical temperature was above core (~37oC), and the opposite warming effect when baseline cortical temperature was below core temperature. **c-e** Open circles denote individual animal data, with each colour representing each modulated temperature condition. Filled diamonds of the same colour indicate average across animals. Dashed grey lines denote 95% confidence bounds of curve fitting (light blue in **c** and **d**, solid grey in **e**) to averaged data. See the main text for curve fitting and statistical details.

ANOVA, 2 s, $F = 5.65$, df $= 6$, $p = 0.00066$; 16 s, $F = 13.02$, df $= 6$, $p = 1.09 \times 10^{-7}$), with lower cortical temperatures observed to be associated with a relative reduction in evoked LFP magnitude and a broadening of the response as compared to warmer cortical temperatures (Fig. 2b, with stronger effects seen for 16 s stimulation). Notably, a striking inverted U-shaped relationship (of the form $y = a + bx + cx^2$, curve fitting implemented using non-linear least squares, goodness-of fit 2 s, 0.92; 16 s, 0.97) between baseline cortical temperature and evoked LFP amplitude was observed, with the greatest response modelled to occur at 31.5 °C (2 s) and 30.8 °C (16 s) (Fig. 2c, note evoked LFP amplitudes as absolute values). Similarly, evoked multi-unit activity (MUA), averaged across stimulation impulses, exhibited marked transient increases across cortical layers (Fig. 2d), with a statistically significant effect of

cortical temperature on evoked granular MUA amplitude (single factor ANOVA, 2 s, $F = 33.38$, df $= 6$, $p = 1.17 \times 10^{-12}$; 16 s, $F = 43.81$, df $= 6$, $p = 7.58 \times 10^{-15}$), and an overall broadening and reduction in peak amplitude with decreasing cortical temperature (Fig. 2e and see also Supplemental Table 3 for comprehensive statistics). Consistent with evoked LFP observations, baseline cortical temperature was again found to be non-linearly related to evoked MUA amplitude (inverted U-shaped relationship of the form $y = a + bx + cx^2$, goodness-of fit 2 s, 0.92; 16 s, 0.98), with maximal responses modelled to occur at 27.9 °C (2 s) and 28.9 °C (16 s) (Fig. 2f). These results indicate there to be a profound non-linear relationship between cortical temperature and sensory-evoked neural responses, with a predicted maximal operating range of 29.8 ± 0.7 °C under our experimental conditions.

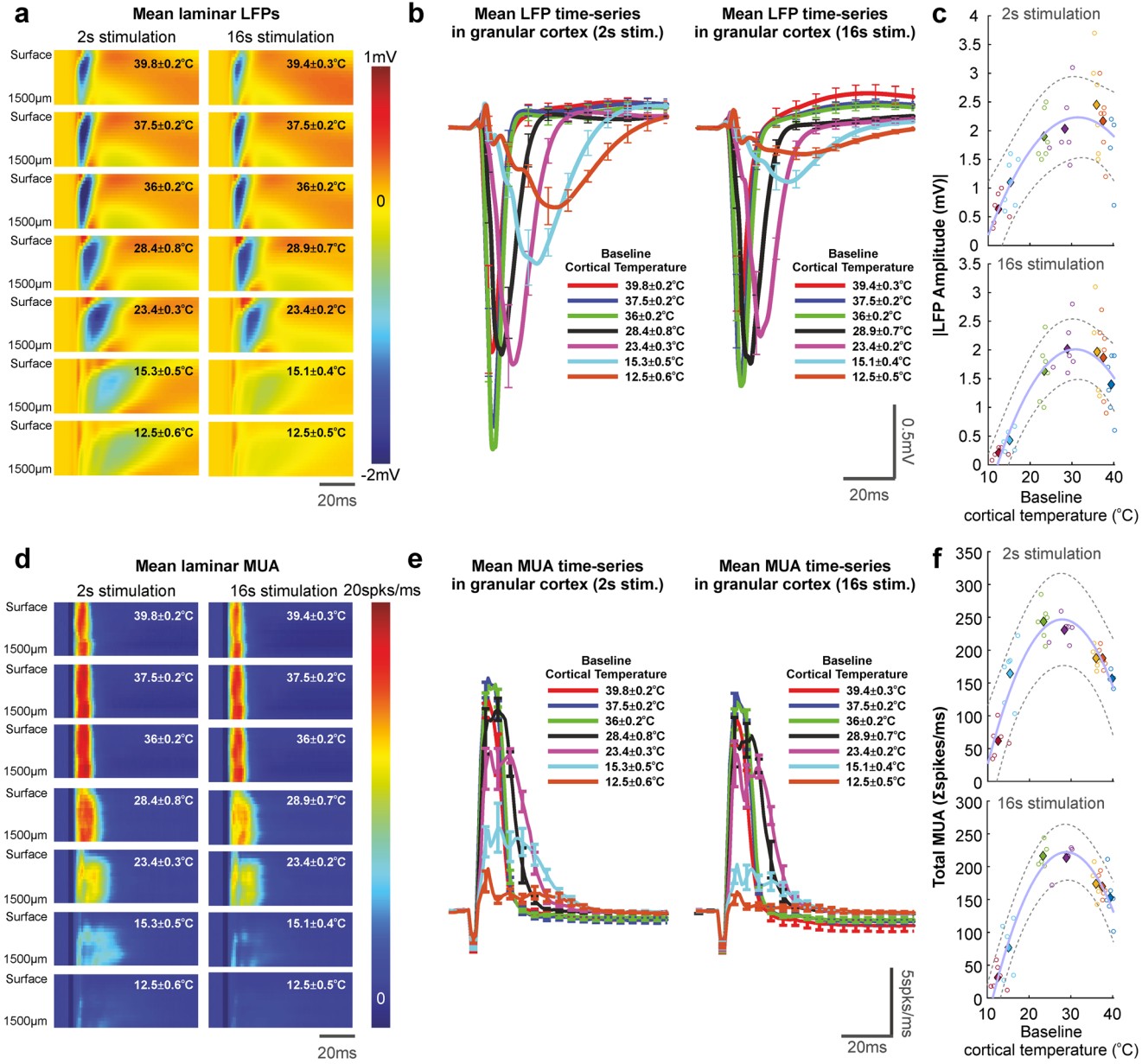

**Fig. 2 Evoked local field potential (LFP) and multi-unit activity (MUA) responses to 2 s and 16 s whisker stimulation in barrel cortex under a range of cortical temperatures. a** Example laminar profile of averaged LFP responses to whisker stimulation as a function of cortical temperature. **b** Mean evoked LFP timeseries in the granular cortex to whisker stimulation. **c** Non-linear relationship between baseline cortical temperature and evoked absolute LFP amplitude. **d** Laminar profile of average MUA responses to whisker stimulation as a function of cortical temperature. **e** Mean evoked MUA timeseries in the granular cortex to whisker stimulation. **f** Evoked MUA responses exhibited a non-linear relationship with baseline cortical temperature as seen in evoked LFP measures. **c, f** Open circles denote individual animal data, with each colour representing each modulated temperature condition. Filled diamonds of the same colour indicate average across animals. Dashed grey lines denote 95% confidence bounds of curve fitting (light blue) to averaged data. See the main text for curve fitting and statistical details.

**Non-linear relationship between cortical temperature and local hemodynamics during sensory processing and evidence for delayed vasodilation in the cooled cortex.** Concurrent spatio-temporal recordings of total haemoglobin (Hbt), oxyhaemoglobin (Hbo), and deoxyhaemoglobin concentration (Hbr) change during 2 and 16 s whisker stimulation (Fig. 3a), at moderate cortical temperatures (~29–37.5 °C), displayed canonical focal onset (at ~1 s post stimulation onset) in barrel cortex with subsequent increase in spatial coverage (Fig. 3b). In contrast, during elevated cortical temperatures, these were relatively slow to peak, reduced in amplitude and more spatially diffuse, and were also intriguingly associated with the emergence of a persistent low-frequency

oscillation that was present in both short and long stimulation periods (Fig. 3b, c, see insets). When examined as a function of cortical temperature, Hbt peak amplitude responses to both 2 and 16 s sensory stimulation were significantly affected by cortical temperature (single factor ANOVA, 2 s, $F = 5.48$, df = 6, $p = 0.0005$; 16 s, $F = 8.959$, df = 6, $p = 6 \times 10^{-6}$) and exhibited an inverted U-shape relationship, akin to that seen in neural measures (see Fig. 2c, f), and which, when modelled (goodness-of fit 2 s, 0.94; 16 s, 0.93), indicated a maximal Hbt response at cortical temperatures of 28.7 °C (2 s) and 27.1 °C (16 s) (Fig. 3c). Moreover, cortical temperature significantly modulated evoked Hbt onset time (single factor ANOVA, 2 s, $F = 4.2$, df = 6, $p = 0.003$; 16 s,

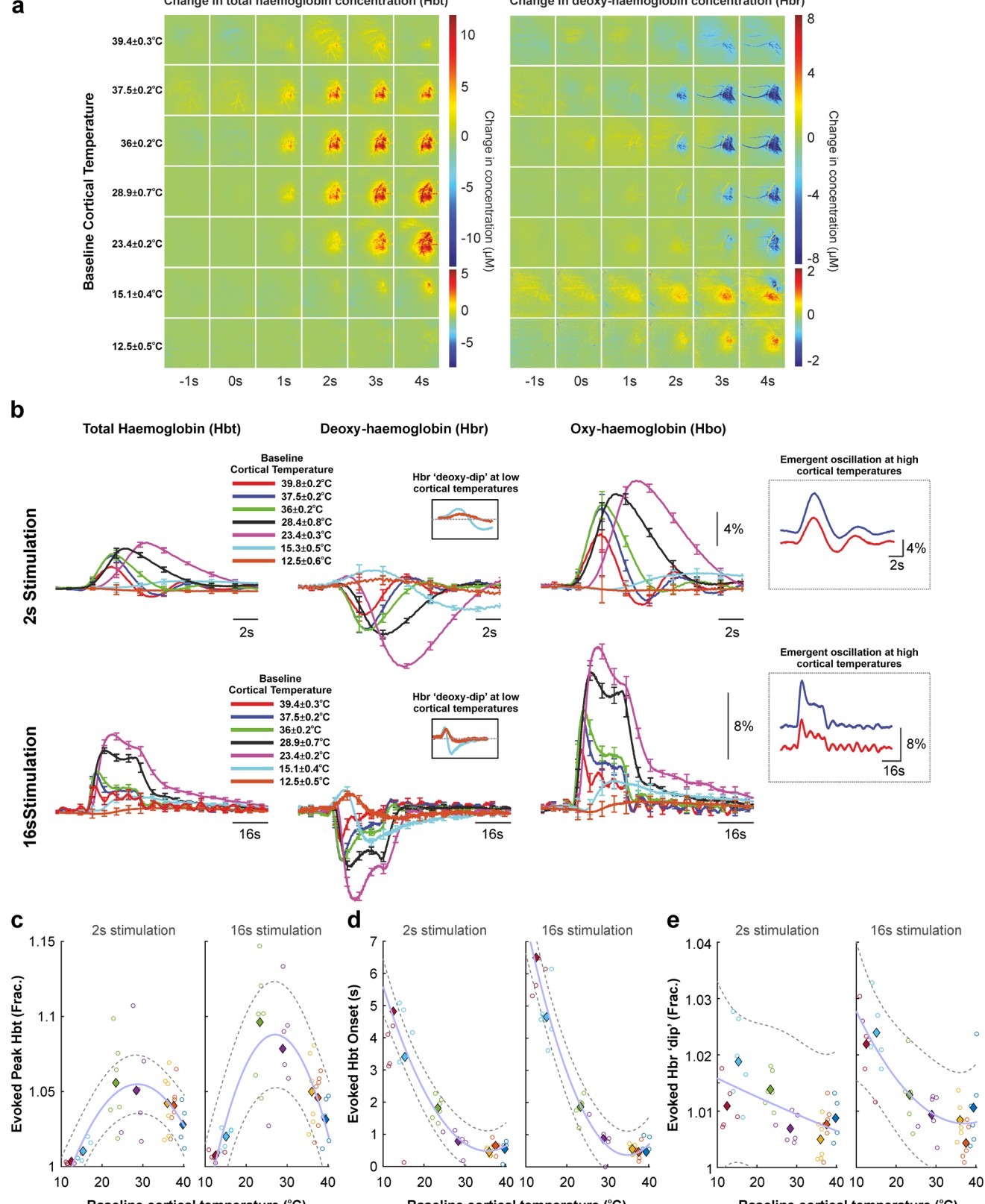

$F = 89.24$, df = 6, $p = 9.8 \times 10^{-20}$), and was manifest as a non-linear monotonically decreasing relationship, such that cortical hypothermia was associated with a pronounced delay of >4 s seconds relative to responses under elevated temperatures (Fig. 3d). Thus, evoked Hbt responses under cool cortical temperatures (12–15 °C) were markedly delayed, diminished in amplitude, and, with respect to Hbr, displayed a surprising early increase in concentration (i.e. a 'deoxy-dip') that was more sustained than would be typically expected due to washout and which was observed at higher cortical temperatures (Fig. 3b, see insets and also Table 3 for comprehensive statistics). Statistical analyses revealed a significant effect of temperature on the magnitude of the 'deoxy-dip'

**Fig. 3 Evoked haemodynamic (total haemoglobin concentration [Hbt], oxyhaemoglobin [Hbo], and deoxyhaemoglobin [Hbr]) responses to 2 s and 16 s whisker stimulation in barrel cortex under a range of cortical temperatures. a** Example spatiotemporal changes in Hbt and Hbr in a single animal during 16 s whisker stimulation. **b** ROI extracted mean Hbt, Hbo and Hbr timeseries during 2 s and 16 s whisker stimulation. Note middle insets indicating an early increase in Hbr ("deoxy-dip") at cool cortical temperatures and right insets illustrating the emergence of a low-frequency oscillation at elevated cortical temperatures. **c** Non-linear relationship between baseline cortical temperature and peak evoked Hbt. **d** Non-linear monotonically decreasing relationship between baseline cortical temperature and evoked Hbt onset. **e** Increased presence of a rise in Hbr ('deoxy-dip') during 2 s and 16 s whisker stimulation with decreasing cortical temperature. **c-e** Open circles denote individual animal data, with each colour representing each modulated temperature condition. Filled diamonds of the same colour indicate average across animals. Dashed grey lines denote 95% confidence bounds of curve fitting (light blue) to averaged data. See the main text for curve fitting and statistical details.

(single factor ANOVA, 2 s, $F = 7.006$, df = 6, $p = 7.2 \times 10^{-5}$; 16 s, $F = 11.63$, df = 6, $p = 4.0 \times 10^{-7}$, see also supplemental Table 3 for comprehensive statistics) and this was manifest as an overall inverse relationship, such that the 'deoxy-dip' magnitude increased with a reduction in cortical temperature (Fig. 3e). Taken together, these results demonstrate that changes in brain temperature dramatically modulate the magnitude and timing of hemodynamic responses in a predominantly non-linear manner during sensory processing.

**Emergence of a low-frequency oscillation at elevated cortical temperatures.** An intriguing observation made when examining evoked haemodynamics responses as a function of cortical temperature was that of a distinct low-frequency oscillation at elevated temperatures (see insets in Fig. 3b). On further analysis, this oscillation was associated with an increase in signal power in the 0.05–0.25 Hz frequency range (Fig. 4a, b) and was augmented as a function of modulated cortical temperature (Fig. 4c). Since burst suppression phenomena also operate at overlapping frequencies, and may manifest under anaesthetic conditions such as those employed here, we examined whether such activity might in some way underpin the observed hemodynamic oscillation at higher cortical temperatures. Calculation of the burst suppression ratio (BSR, see Methods) in LFP data indicated an increase in burst suppression with decreasing cortical temperature, as expected from previous reports employing therapeutic hypothermia[28] (Fig. 4d). Nevertheless, in five of six animals in which a clear low-frequency oscillation was observed at the most elevated cortical temperature (~39 ºC), differences in BSR were linearly correlated to the presence of the aforementioned hemodynamic oscillation, albeit with a positive y-axis intercept (Fig. 4e), suggesting (together with the observation that burst suppression at lower cortical temperatures was not associated with the oscillation) that other factors beyond burst suppression effects contribute to the emergence of the low-frequency oscillation under hyperthermia. To further illustrate this effect, we selected the maxima of Hbt oscillations (±10 s window) during the first ~300 s of each experiment which resulted in an averaged haemodynamic trial with Hbt peaking at time zero, and which could be subsequently compared to laminar MUA extracted over the same time window. When examining experiments during which burst suppression was particularly evident (Fig. 4f), Hbt oscillatory maxima were preceded by ~2 s by a transient MUA increase in granular and infragranular cortical layers, a lag that is comparable to stimulation-evoked neurovascular coupling. In turn, in another experiment that also displayed a robust baseline oscillation in Hbt, albeit with no evidence for burst-suppression, no changes in baseline neuronal activity could be discerned (Fig. 4g). While correlative in nature, this analysis suggests that the low-frequency oscillation observed at elevated cortical temperatures may emerge from a neuronal-independent mechanism that may thus be of utility as a marker for cerebral hyperthermia using non-invasive imaging techniques.

**Non-linear neurovascular coupling and evidence for reduced tissue oxygen during sensory processing in the cooled cortex.** We next examined stimulus-evoked changes in tissue oxygenation ($pO_2$) and how these related to concurrent haemodynamic measures. We found there to be a non-linear relationship (again well characterised by an inverted U-shaped function of the form $y = a + bx + cx^2$, goodness-of fit 2 s, 0.82; 16 s, 0.89) between baseline cortical temperature and evoked changes in $pO_2$ (Fig. 5a), with cortical temperatures under $15.6 \pm 0.3$ ºC (pooled stimulation conditions) being associated with a decrease in evoked $pO_2$ below baseline, indicative of transient hypoxia during sensory stimulation (see shaded regions in Fig. 5a). This observation was consistent with a tight negative correlation ($r \leq -0.82$, $p \leq 0.02$, both stimulation conditions) between evoked $pO_2$ and the peak magnitude of the Hbr 'deoxy-dip', such that transient evoked hypoxia at low cortical temperatures was associated with an increase in evoked Hbr concentration (Fig. 5b). Furthermore, evoked $pO_2$ was also negatively correlated to the onset of evoked Hbt ($r \leq -0.86$, $p = 0.01$, both stimulation conditions), in which transient evoked hypoxia at low cortical temperatures was coupled to an increased delay (2–3 s) in Hbt onset relative to warmer cortical temperatures (Fig. 5c). Finally, we examined neurovascular coupling during cortical temperature manipulation through comparison of peak evoked LFP and Hbt responses to 2 s and 16 s whisker stimulation (Fig. 5d). This indicated a non-linear relationship (again well characterised by an inverted U-shaped function of the form $y = a + bx + cx^2$, goodness-of fit 2 s, 0.93; 16 s, 0.95) such that lower cortical temperatures were associated with a steeper increase in evoked LFP activity, relative to Hbt, in contrast to warmer cortical temperatures (Fig. 5d). Taken together, these observations indicate that changes in cortical temperature profoundly impact the timing and magnitude of evoked functional hyperaemic responses, whereby brain cooling below ~15 ºC, in particular, leads to a scenario where tissue oxygen is extracted prior to vasodilation, causing transient tissue hypoxia, and which results in a robust early increase in Hbr (i.e. 'the deoxy-dip').

**Discussion**

In summary, we employed a novel multi-modal methodology, alongside precise manipulation of cortical temperature, to interrogate the spatiotemporal dynamics of multiple neural and vascular measures during sensory processing, and the role of brain temperature in modulating these. We demonstrate that changes in cortical temperature profoundly modulate sensory-evoked neural and vascular responses in a non-linear (partially inverted U-shape) manner, where, notably, brain cooling substantially dampens neural and vascular responses and delays neurovascular coupling, such that oxygen is extracted before the vessels begin to dilate and leading to the emergence of transient tissue hypoxia and a robust early increase in Hbr (the 'deoxy-dip'). In turn, elevating brain temperature above the core was associated with moderate blunting and faster dynamics of neural and vascular responses, as well as the emergence of an intriguing, but robustly observed, low-frequency oscillation in hemodynamic measures.

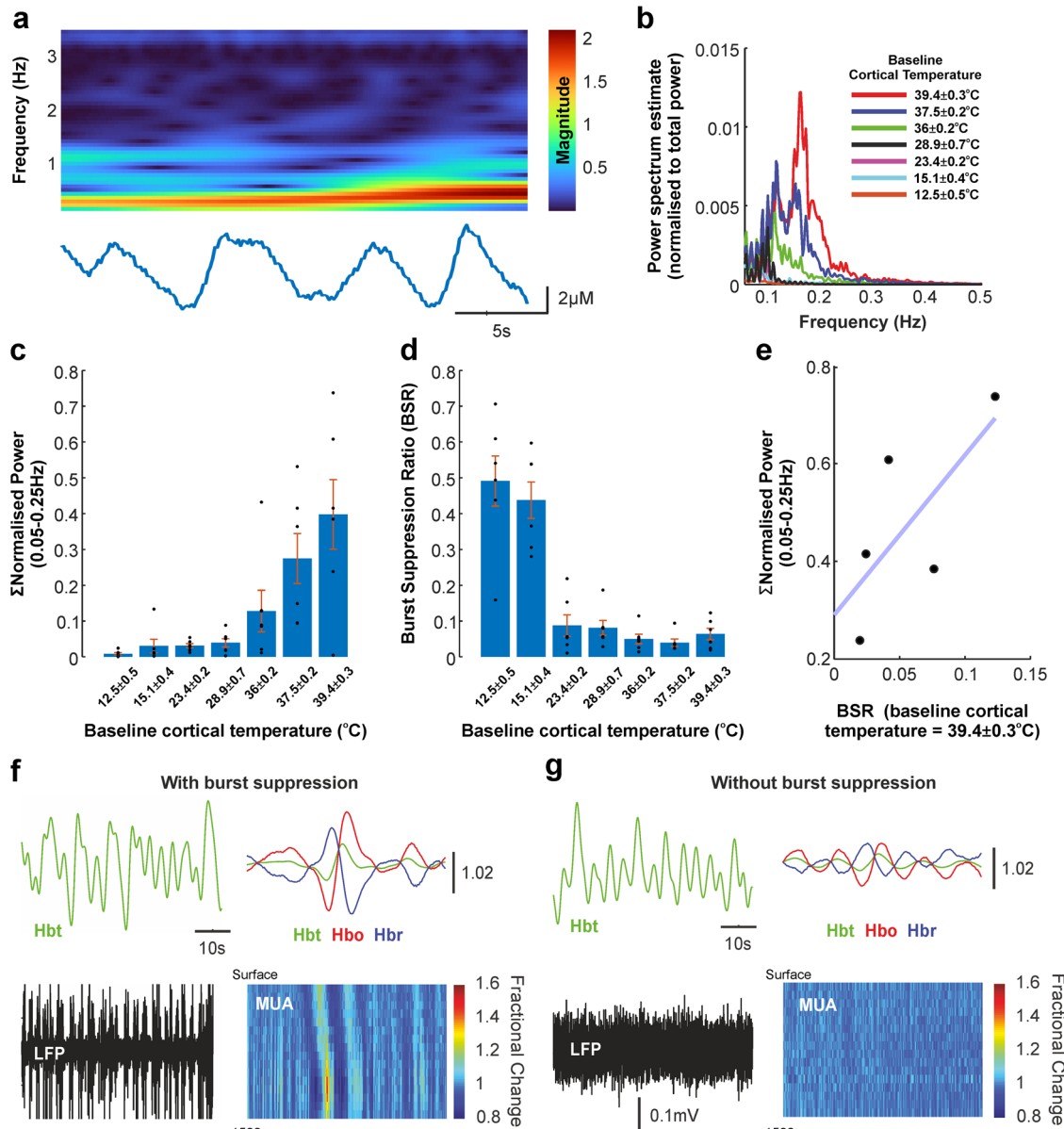

**Fig. 4 Emergence of a low-frequency oscillation at elevated cortical temperatures. a** Continuous wavelet transform of a sample inter-stimulus period in which a low-frequency oscillation with range ~0.05–0.25 Hz can be clearly discerned in the Hbt timeseries. **b** Normalised Welch's power spectrum estimate (0.05–0.25 Hz) of concatenated experimental Hbt timeseries (16 s whisker stimulation condition) across the range of cortical temperatures studied (see key) averaged across animals (error bars omitted for clarity, see quantification in **c**). **c** Quantification of normalised data in (**b**) summed across animals ($N = 6$), indicating an increase in power in the frequency range 0.05–0.25 Hz with increasing cortical temperature. **d** Burst suppression ratio (BSR) across animals ($N = 6$), a measure of the prevalence of burst suppression phenomena in LFP timeseries, was found to be highest at cooler cortical temperatures (consistent with previous reports) and to decrease with increasing cortical temperature. **e** BSR and normalised power in the frequency range associated with the observed low-frequency oscillation at the most elevated cortical temperature studied, in 5/6 animals which displayed burst suppression, was found to be strongly correlated albeit with a positive y-axis intercept, suggesting that burst suppression does not solely underpin the emergence of the pathological oscillation. **f**, **g** Corroboration of interpretation from (**e**) in two contrasting example animals, in which the pathological oscillation is manifest in the presence (**e**) and absence (**f**) of burst suppression, the former being associated with an increase in infragranular MUA at physiological timescales for neurovascular coupling.

Perfusion-related imaging techniques, such as BOLD fMRI, do not provide a readout of underlying changes in neuronal activity, but rather provide hemodynamic surrogate measures as a result of (assumed linear) neurovascular coupling. This emphasises the criticality of a complete mechanistic understanding of this biological process for accurate interpretation of such neuroimaging signals. An initial observation in the field was that of an early increase in deoxyhaemoglobin (Hbr), a 'deoxy-dip', during sensory processing which promised to map active regions in the brain more

accurately, since it would be expected to produce an early focal negative BOLD signal prior to subsequent Hbr washout (which underpins the canonical positive BOLD signal due to functional hyperaemia) in draining veins spatially removed from the site of neuronal activation[29]. However, the validity of the 'deoxy-dip' remains hotly contested, with groups including our own reporting its presence[30–33] and others failing to observe it[34], with the suggestion it might be an artefact of the optical imaging algorithms used to convert remitted light into changes in haemoglobin

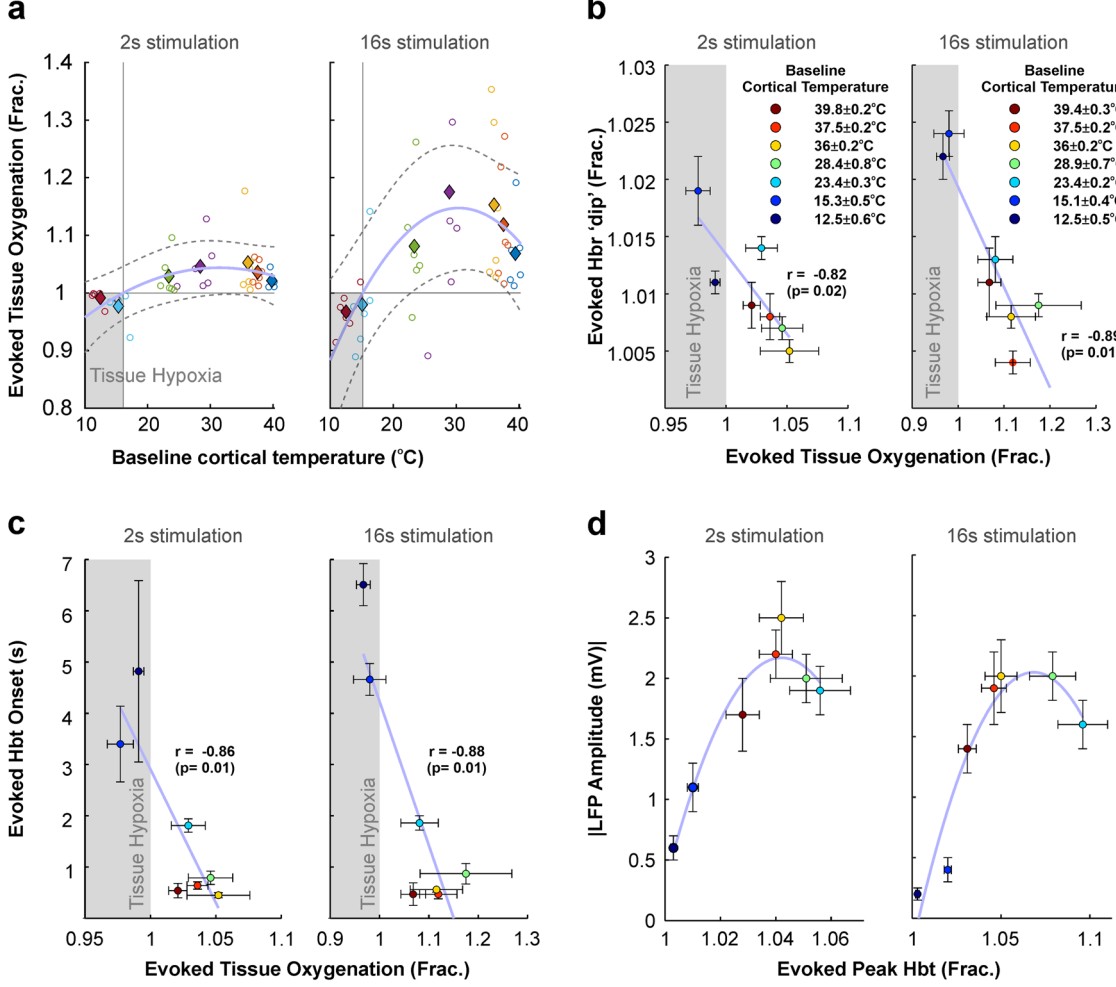

**Fig. 5 Evaluation of oximetry, hemodynamic responses, and neurovascular coupling to whisker stimulation under a range of cortical temperatures.**
**a** Non-linear relationship between changes in evoked tissue oxygenation as a function of cortical temperature to 2 s and 16 s whisker stimulation. Open circles denote individual animal data, with each colour representing each modulated temperature condition. Filled diamonds of the same colour indicate average across animals. Dashed grey lines denote 95% confidence bounds of curve fitting (light blue) to averaged data. **b** Negative linear correlation between changes in evoked tissue oxygenation and magnitude of the increase in evoked Hbr concentration ('deoxy-dip') across cortical temperatures. **c** Negative linear correlation between changes in evoked tissue oxygenation and onset time of evoked Hbt response. **d** Non-linear relationship between evoked LFP responses to 2 s and 16 s whisker stimulation, and magnitude of peak evoked Hbt response. **a–c** Shaded areas indicate cool cortical temperatures during which sensory stimulation induces a transient period of hypoxia. **b–d** Data points represent averages across animals with xy-error bars as SEM, and colour coded to each modulated temperature condition as given in key in **b**. See the main text for curve fitting and statistical details.

concentration[35,36]. Our results indicating that cooler cortical temperatures are associated with the presence of a 'deoxy-dip' potentially reconcile this historic debate in the field, and suggest that the variable findings may have arisen due to differing cortical temperatures as a result of different experimental methodologies, e.g., exposed dura versus thinned cranial window preparations, laboratory environment, anaesthetic regime and duration[37], and even the water for immersion objectives[38].

Regarding therapeutic implications, focal cooling of the brain to similar temperatures as employed here reduces the risk of epileptic activity during surgery which may manifest when conducting intraoperative functional mapping using electrical stimulation methods[39]. However, intraoperative focal cooling of specific brain regions during the performance of vocal sequences in awake neurological patients was found, in itself, to enable functional segregation of cortical areas underpinning speech timing and articulation, and aid avoidance of critical speech centres during surgical resection[40]. It has also recently been reported that intraoperative direct focal cooling of the brain,

using a similar PID-controlled system approach and target temperature as in our study, enabled language mapping through monitoring of thermal changes due to functional hyperaemia[41] (and see Fig. 1e in the current manuscript). Since intrinsic optical imaging spectroscopy (OIS) methods, identical to those used here, have also been successfully implemented in the operating theatre[42–45], and given our observation of a 'deoxy-dip' during sensory processing at cool cortical temperatures that is spatially localised to the site of neuronal activation (see above), it is thus tempting to speculate that the combination of focal cooling and OIS intra-operatively would not only reduce the risk of epileptiform activity during associated functional mapping, but also improve mapping accuracy.

In turn, we made an intriguing observation of a persistent low-frequency oscillation (0.05–0.25 Hz) in hemodynamic measures with cortical temperatures substantially above core levels. Cerebral vasomotion is a long observed, but increasingly appreciated, vascular oscillation[46–48], the aetiology and function (or indeed characteristics) of which remain debated, but which in rat-derived

hemodynamic data takes the form of a narrowband peak centred on 0.1 Hz on the frequency spectrum under normal experimental conditions[48]. The observed oscillation may therefore reflect a distinct brain hyperthermia-related oscillation, or perhaps a higher frequency variant of vasomotion; in either case, it remains to be confirmed whether this emergent signal reflects pathological processes as a result of elevated cortical temperature, or confers some form of neuroprotection as has been posited for vasomotion, see ref. [49]. Correlation analysis suggested that the observed low-frequency oscillation may comprise a neural-independent component and also linked to burst suppression activity, however, this requires further investigation. Notwithstanding, functional near-infrared spectroscopy (fNIRS) is a technique that shares an analogous biophysical framework as OIS (as employed here, simply at relatively shorter wavelengths), and in light of the value of fNIRS for non-invasive neonatal/infant monitoring[50], it is again tempting to speculate that the observed oscillation may be a useful functional marker of brain hyperthermia during development that is non-invasively detectable by fNIRS, particularly since pathological brain hyperthermia may not be faithfully reflected in measures of core body temperature[51].

Beyond revealing an inverted U-shaped curvilinear relationship between cortical temperature and neurovascular coupling, suggesting an optimal temperature window for function and with implications for the interpretation of BOLD fMRI signals (see also our previous work[17]), we made the intriguing observation that maximal sensory-evoked neural and hemodynamic responses occurred at cortical temperatures of approximately 29 °C (Figs. 2 and 3). The significance of this effect remains unclear, but may occur due to rodents possessing brains with high surface-to-volume ratios that are particularly susceptible to heat exchange with the external environment, and which exhibit negative brain-core temperature differentials in contrast to humans[52]. Mechanistically, moderate cortical temperatures were associated with a broadening of the neural response and suggestive of a reduction in feedforward inhibition. At elevated temperatures, the reverse was true, with neuronal firing rapidly curtailed, and suggestive of enhanced neuronal inhibition. In this context, it is interesting to note that cortical interneurons may be more adversely affected by temperature variations relative to excitatory neurons/excitation[53,54], and play a key role in shaping BOLD signal dynamics[55–57]. It will thus be interesting for further research to uncover whether other cells that form part of the neurovascular unit, namely astrocytes and pericytes, are also similarly susceptible to changes in cortical temperature.

**Conclusion and methodological considerations**. Pharmacological manipulations to investigate neurovascular coupling have made important contributions to our understanding of mechanistic pathways and cell types involved in this critical homoeostatic process. Here, through brain temperature modulation, we present an alternate elementary, but the granular, method to systematically alter evoked cortical processing and assess its effects on neurovascular coupling which circumvents potential off-target effects. Using this approach, we reveal that bidirectional changes in cortical temperature markedly alter neural and vascular responses during sensory processing. Our findings have important implications for interpreting functional data from in-vivo experiments in which local brain tissue temperature is altered beyond normothermia by the techniques employed (e.g., optogenetic stimulation, laser illumination, and anaesthetic regime)[37,38,58,59]. Furthermore, our results may inform burgeoning clinical methods utilising focal brain cooling at temperatures studied here, including for chronic and intraoperative suppression of epileptic activity and functional mapping[39–41,60,61], as well as providing insights into potential pathological processes in neurological conditions where brain temperatures exceed core levels by up to several °C[62–64]. Notwithstanding, our data is derived from the anaesthetised rodent due to the invasive nature and technical challenges involved in these temperature modulation experiments, and include thermal extrema which would not be experienced under normal physiological conditions. Factors such as anaesthesia-related confounds and differing intrinsic brain thermodynamics in rodents versus humans are thus important considerations when interpreting our results. While at present it is not practicable to recapitulate such experiments in humans, technological developments may enable this during intraoperative procedures in suitable cohorts in future and allow validation of our pre-clinical results.

## Methods
Surgical and experimental procedures were regulated by the UK Home Office, in accordance with the Animal (Scientific procedures) Act 1986. Six female Hooded Lister rats weighing between 230 to 330 g were kept in a temperature-controlled environment (20–22 °C) under a 12-h dark/light cycle. Food and water were supplied ad libitum. Animals were anaesthetised with urethane at 1.25 g/kg i.p. with additional doses (0.1 ml) administered if required. Urethane anaesthesia has been shown to preserve both excitatory (glutamate-mediated) and inhibitory (GABA-A-and GABA-B-mediated) synaptic transmission. This is in contrast to many general anaesthetics, which are thought to enhance GABAergic and/or inhibit glutamatergic transmission[65–68]. Experiments were conducted under terminal anaesthesia due to the invasive nature of the temperature-modulation approach, as well as the possibility of inducing febrile seizures during elevated cortical temperatures where animals were conscious. Atropine was administered at 0.4 mg/kg s.c. to lessen mucous secretions during surgery. Core temperature was maintained at ~37 °C throughout surgical and experimental procedures using a homoeothermic heating blanket system with rectal temperature monitoring (Harvard Instruments, Edenbridge, UK). Animals were tracheotomised to allow controlled ventilation and continuous monitoring of end-tidal $CO_2$ (CapStar-100, CWE Systems, USA). Systemic $pCO_2$ and oxygen saturation were maintained within physiological limits through the adjustment of ventilator parameters, according to arterial blood gas measurements. The femoral vein and artery were cannulated to allow measurement of systemic arterial blood pressure (MABP) and drug infusion respectively. MABP was maintained between 100–110 mmHg through an infusion of phenylephrine at 0.13–0.26 mg/h[69,70]. A stereotaxic frame (Kopf Instruments, California, USA) was used to hold the subject. The skull was exposed and a region of the skull overlying the right somatosensory cortex was thinned to translucency using a dental drill. Sterile saline was used to cool the cortical surface during drilling. A circular plastic chamber and coil for cortical temperature modulation (described below) was then secured to the skull using dental cement.

**Electrode and temperature/oxygen sensor placement**. In order to place electrodes and temperature/oxygen sensors into the whisker barrel cortex we first performed short 2 s whisker stimulation to localise the active region. Two stainless-steel stimulation electrodes, insulated to within 2 mm of the tip, were inserted subcutaneously in an anterior-to-posterior plane into the left (contralateral) whisker pad. The electrodes were placed between rows A/B and C/D, respectively; to ensure the entire whisker pad was stimulated when electrical pulses were applied. These electrical stimuli produced no changes in measured systemic physiology (MABP, heart rate or end-tidal $pCO_2$). The 2 s whisker experiment was performed with 2D-OIS and consisted of 30 trials of 25 s duration with a 2 s stimulation at 5 Hz with a 5 s baseline period in each trial. The data were then averaged and subjected to spectroscopy analysis described below to produce micromolar changes in Hbt over time.

Images were analysed to produce a temporally averaged spatial map of Hbt during stimulation with automated image segmentation analysis to find the boundary surrounding the primary region exhibiting an increase in Hbt. The boundary was overlaid onto a camera reference image showing the surface vasculature (See Fig. 1b). Regions in the centre of the activated whisker barrels with minimal surface vasculature were selected for placement of the probes to prevent bleeding on implantation. The electrode location was prioritised over the temperature/oxygen probe. A small hole was drilled in the remaining skull overlying each selected region and the dura was pierced using a fine needle. The 16-channel linear array electrode (177 $\mu m^2$ site area, 100 $\mu m$ spacing, Neuronexus Technologies, USA) was attached to a stereotaxic arm (Kopf Instruments, USA) and inserted normal to the cortical surface to a depth of approximately 1500 $\mu m$. A multi-probe (for measurement of tissue oxygenation and temperature, see below) was also attached to a secondary stereotaxic arm and inserted normally to the cortical surface to a depth of 500 $\mu m$ using stereotaxic manipulation under a microscope (see Fig. 1b for examples of electrode and probe placed into the whisker barrel cortex). Sterile saline was then placed in the well and covered with a partial glass coverslip to provide stable imaging throughout all procedures and changes in chamber fluid temperature.

**Cortical temperature control**. A novel system was developed to modulate the cortical surface temperature of an anaesthetised rat with fine control (Fig. 1a). The system used two fluid-filled reservoirs to supply either heated (50 ºC) or cooled (<1 ºC) distilled water. The heated water component employed a temperature-controlled hot water bath (StableTemp 5 litre, Cole-Parmer UK) set at 50 ºC. The cooled water component consisted of a 3-l plastic container filled with a 50/50 mix of ice and distilled water and placed inside a polystyrene box for insulation. Both reservoirs contained submerged thermocouples to monitor the temperature. Both reservoirs were allowed to stabilise for at least an hour before active temperature control, and were topped up with small volumes (<10% of the total volume) of fresh distilled water when required. The temperature of both reservoirs remained stable throughout experiments. Micropumps (M100S-SUB, TCS Micropumps, Kent UK) were submerged into both reservoirs, with the outlet of each pump attached to small bore flexible silicone tubing that was subsequently attached to a larger bore silicone tubing (6 mm OD, 4 mm ID, Aquarium tubing). The outlet tubes from the two pumps were insulated and connected to a T-piece ~70 cm from each reservoir that provided inflow into the skull-attached chamber. The internal circumference of the chamber contained a stainless-steel tube which was formed into a coil which provided cooling and heating to a sterile saline reservoir inside the chamber. The inlet to this coil was attached to PVC tubing and then to the combined inflow from the hot and cool reservoirs (see above). The outlet was also attached to PVC tubing and secured into a drain vessel to capture wastewater. The top of the chamber contained a machined lip so as to hold a microscope glass coverslip, with room for the multi-channel electrode, brain temperature/oxygen sensor and a separate thermocouple to measure the temperature of the saline reservoir inside the chamber to allow the performance of the heating/cooling circuit to be continually monitored.

Our method to control brain temperature produced remarkably stable results. We performed whisker stimulations (2 and 16 s) over a range of set chamber fluid temperatures (range 6–44 ºC) which modulated cortical temperatures from 12.5 to 39.8 ºC (see Supplementary Tables 1, 2 for further details). The difference between chamber temperature and brain temperature reflects the dynamic nature of the brain where the influx of core-temperature blood provides either a cooling or heating effect depending on brain temperature.

**Controller design**. Fine control of temperature is often difficult due to the inherent time delays common in temperature systems. The control problem was exacerbated here by a small volume of fluid to be controlled, a wide range of required set-points (6 to 44 ºC) above and below the room and subject's temperature, fast transition times required when changing between temperatures and minimal steady-state error. Thus, a multi-step approach was required with fine control of fluid flow. The top-level control software and user interface was written in LabVIEW (National Instruments, Texas USA) which allowed for bidirectional communication between the controller PC and the temperature sensing and pump controller. Temperatures were recorded using thermocouples (Type K, Pico Technology) submerged in the chamber, as well as hot and cold-water reservoirs. These were connected to a data logger (TC-08, 8 Channel data logger, Pico Technologies, Cambridgeshire), which sampled up to 1 kHz and was connected to the PC via USB. A software development kit (SDK) was used to interface the data logger with LabVIEW (PicoSDK 10.6.12, Pico Technology).

**Pump control**. The pump control output from LabVIEW was linked to a programmable microprocessor (Arduino UNO R3, Arduino Italy) via a custom RS232 interface that allowed for programmatic selection of pump (hot or cold) and pumps speed within a range of −255 (most cold) to +255 (most hot). A proportional-integral derivative (PID) control system was developed in LabVIEW. PID offers continuous system control through the adjustment of a manipulated variable (MV), to minimise an error value arising from the difference between the desired value (set-point, SP) and a measured value (process variable, PV). Here, SP is the desired temperature (6 to 44 ºC), PV is the temperature measured from the thermocouple submerged in the well, and MV is the pump source and power. The three components of the PID offer different aspects of control that can be individually tuned to match the system being controlled and the performance characteristics required. Proportional, proportional-integral and PID controllers were implemented and manually tuned, but were unable to maintain a stable well temperature. Thus, further control steps were incorporated to enhance the controller function. The first, added the low-pass filtering of the PV input to a derivative component of the controller to reduce noise effects from PV temperature sampling. The second, PID gain scheduling, was included to alter controller gain parameters when the temperature was within 1 ºC of SP. This allowed more aggressive control of temperature when large changes in SP were required, while more slow-acting parameters were used when the error term was minimal, e.g., SP near PV. The hardware design also facilitated more stable software control, especially during large changes in SP, such as the minimal pumping, insulated pipework, one-way check-valves and minimal tube length between the t-piece and well. The cooling control hardware (Arduino) received 5 v TTL pulses from the CED data acquisition hardware for every stimulus presentation onset, which was passed via RS232 into the LabVIEW. The synchronisation pulses were recorded alongside the recordings of well temperature and process control parameters, allowing later comparison with the other multi-modal neuroimaging methods. The resultant

temperature control system ensured stability at constant temperature and a rapid change (less than two minutes) to the new set temperature when required.

**Two-Dimensional optical imaging spectroscopy**. Two-Dimensional optical imaging spectroscopy (2D-OIS) provides spatiotemporal measures of cortical haemodynamics. The surface of the cortex was illuminated with four wavelengths of light in a repeating sequence, with a CCD camera (1M60, Teledyne Dalsa, Canada) used to record the light remitted. The camera operated with 4 × 4 pixel binning, with each image pixel representing 75 × 75 μm of the cortical surface. The quantum efficiency of the camera was 28% at 500 nm. Illumination and wavelength switching was provided by a Lambda DG-4 high-speed filter changer (Sutter Instrument Company, Novato, California, USA). The four wavelengths were chosen as two specific pairs (494 nm ± 31 FWHM and 560 nm ± 16 FWHM; 575 nm ± 14 FWHM and 595 nm ± 9 FWHM). The wavelengths selected for each pair had similar total absorption coefficients and thus sampled the same tissue volume. The specific absorption coefficients for oxyhaemoglobin (HbO2) and deoxyhaemoglobin (Hbr) were chosen to be as different as possible for each pair, to maximise the signal-to-noise ratio. The 32 Hz frame rate of the camera was synchronised to the filter switching and thus the four wavelengths of illumination produced an effective frame rate of 32/4 = 8 Hz.

The acquired images were analysed to estimate the change in haemoglobin saturation and concentration from pre-determined baseline values of 100 μM concentration and 60% saturation at ambient room temperature, based on our previous observations[71]. We used the transition period between experiments, during which chamber temperature was modulated to the desired level, to record and calculate the change in baseline Hbt concentration and saturation (see Table below) in order to adjust for temperature-related changes. These values were then used as inputs for spectroscopic analysis for different temperature experiments.

| Chamber temperature (ºC) | Hbt concentration (μM) | Saturation (%) |
|---|---|---|
| 44 | 107 | 59 |
| 40 | 103 | 59 |
| 37 | 102 | 59 |
| Ambient | 100 | 60 |
| 20 | 96 | 66 |
| 10 | 92 | 80 |
| 6 | 95 | 85 |

The analysis approach used the wavelength-dependent absorption spectra of haemoglobin, with estimates of photon path-length using Monte-Carlo simulations of light passing through a homogeneous 3D tissue model, to estimate photon absorption for each illumination wavelength. Images were analysed on a pixel-by-pixel basis using a modified Beer–Lambert law to convert the computed absorption into 2D spatiotemporal image series of the estimates of the changes in total haemoglobin (Hbt), oxyhaemoglobin (HbO) and deoxyhaemoglobin (Hbr) from the baseline values. Data were shown as averaged image maps (in absolute μM change in concentration), or as timeseries from a barrel cortex associated region of interest (ROI) as fractional changes from baseline or in absolute μM change in concentration. Hbt timeseries data were also examined in the frequency domain using a continuous wavelet transform (analytic Morse wavelet with a symmetry parameter of 3 and time-bandwidth product of 60) and Welch power spectrum estimates. Haemodynamic onset times were calculated using a standard approach by identifying 15 and 85% of the peak response and fitting a straight line using linear least squares between these points during the rising phase; where the linear function crossed the baseline was defined as the onset of the hemodynamic response.

**Multi-channel electrophysiology and multi-sensor temperature and oxygen probe**. The electrode was connected to a preamplifier (Medusa Bioamp, Tucker-Davis-Technologies, USA) and optically linked to a data acquisition system (RZ5, Tucker-Davis-Technologies, USA). Data were acquired from 16 channels at 16 bit with 24.4 KHz temporal resolution. Stimulus triggers were recorded by the data acquisition system from the main stimulus control hardware, using TTL pulses, to precisely synchronise equipment. The multi-probe (NX-BF/OT/E pO2 and temperature sensor, Oxford Optronix UK) was connected to an oxygen and temperature monitoring system (OxyLite Pro, Oxford Optronix UK) which outputs tissue oxygenation measures (pO2) and temperature with 1 Hz temporal resolution. The monitoring system was connected to a data acquisition system (CED1401, Cambridge electronic design, UK), which continuously recorded pO2 and temperature throughout experiments. Following confirmation of good subject physiology and probe placement, experimental recordings were commenced after 30 min to allow stabilisation. Data acquired from the electrode and multi-probe were converted and analysed in MATLAB (MathWorks, USA) using custom scripts.

**Local field potential (LFP) analysis**. All stimulation trials were averaged together to create a mean neural trial from each stimulation experiment. LFP data, which represents synaptic activity and intracortical processing, was displayed across all

electrode channels or as a timeseries averaged from depths 400–900 µm (channels 5:10) in order to show the responses from the granular layers in the barrel cortex. To quantify the extent of burst suppression activity for each temperature condition, we extracted the entire concatenated LFP timeseries from depths 400–900 µm, mean-subtracted and averaged and full-wave rectified the signal. The signal was then convolved with a 0.025 ms Gaussian function, with suppression periods recognised as those longer than 0.15 s during which the absolute voltage change did not exceed 25 µV. The total time spent in a suppression state during the experiment was then calculated as a fraction of the total experimental time to provide the burst suppression ratio (BSR).

**Multi-unit analysis**. Raw LFP data was high pass filtered above 300 Hz to remove low-frequency signals. The data were separated into 1 ms temporal bins (each containing 24 samples) and multi-unit activity (MUA) was identified as a threshold crossing 1.5 standard deviations above the mean baseline. Results are displayed as spikes/per millisecond either as a function of all depths recorded, or as an averaged timeseries from depths 400–900 µm (channels 5:10) in order to present data from granular layers.

**Experimental protocol**. Once electrodes and sensors were positioned, and the subject allowed to stabilise for at least 30 min, two stimulation experiments were performed at each set temperature; a short 2 s whisker stimulation (5 Hz, 0.8 mA, pulse width 0.3 ms, 30 trials, 25 s duration with 5 s baseline period) and a prolonged 16 s (5 Hz, 0.8 mA, pulse width 0.3 ms, 30 trials, 96 s duration with 10 s baseline period) whisker stimulation. As well as acquiring data throughout stimulation protocols at different chamber temperatures, data was also acquired during transition periods between the different chamber temperatures in order to allow changes in baseline to be calculated. Chamber temperature was altered sequentially in the following order in order to allow changes in baseline haemodynamic metrics to be calculated from starting ambient conditions (for which there is existing baseline data on haemoglobin concentration and saturation, see above), maintain consistency, and to conduct recordings at elevated temperatures at the end of each animal experiment during which cellular damage might be induced: Chamber temperature = Ambient → 20 °C → 10 °C → 6 °C → 37 °C → 40 °C → 44 °C.

**Reporting summary**. Further information on research design is available in the Nature Portfolio Reporting Summary linked to this article.

# Data availability

Datasets used in the current study are available in the DRYAD repository, https://doi.org/10.5061/dryad.n2z34tmzq.

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

## Acknowledgements

This work was supported by the UK Medical Research Council (Grant No. MR/M013553/1, J.B., L.W.B.) and Epilepsy Research UK (Grant No. P1501, J.B., S.S.H.). S.S.H. is supported by the UK Dementia Research Institute which receives its funding from DRI Ltd, funded by the Medical Research Council, Alzheimer's Society and Alzheimer Research UK. SSH is further supported by a UK DRI Pilot Studies programme award. C.H. is funded by a Sir Henry Dale Fellowship jointly funded by the Wellcome Trust and the Royal Society. This research was funded in whole, or in part, by the Wellcome Trust [Grant number 105586/Z/14/Z]. For the purpose of Open Access, the author has applied a CC BY public copyright licence to any Author Accepted Manuscript version arising from this submission. We thank the technical staff of the University of Sheffield's Department of Psychology, in particular Michael Port for assistance in the construction of the temperature control system.

## Author contributions

All authors contributed to the study's conception and design. Material preparation, data collection and analysis were performed by L.W.B., S.S.H. and J.B. The first draft of the manuscript was written by J.B. and S.S.H., and O.S., L.L., B.E. and C.H. provided critical review, commentary and revisions on all versions of the manuscript. C.H. and J.B. provided resources and administered the project. All authors read and approved the final manuscript.

## Competing interests

The authors declare no competing interests.
