## [Peer Review File · Communications Biology]

Reviewers' comments:

Reviewer #1 (Remarks to the Author):

The manuscript entitled "Bidirectional alterations in brain temperature profoundly modulate spatiotemporal neurovascular responses in-vivo: implications for theragnostics" by Boorman and colleagues is an interesting work that showed differential effects of low and high cortical temperature on oxygen delivery. It provides needed observations on regional thermal effects on NVC and offers possible explanation on the discordance of BOLD data. Below are some of my comments:

1. Did you try to mechanically stimulate the whisker? If you did, did you see any differences in the responses in comparison to whisker pad stimulation?
2. Since temperature changes affect both spatial and temporal responses, have you tried to examine whether low and high temperature has an effect on the central-surround phenomena?
3. It's interesting that the Hbt onset time appeared to be different between 2s stimulation and 16s stimulation at low temperature. What are your thoughts on these differences?
4. There is a "plateau" period after peak response seen with 16s stimulation and that period appeared to be diminished at a relative colder temperature (23-28oC). Would you care to discuss a bit about this?

Reviewer #2 (Remarks to the Author):

Boorman et al. present an approach to experimentally modulate cortical temperature while simultaneously measuring neuronal and vascular dynamics. The authors convincingly demonstrate the ability to finely change cortical temperature (Fig 1C) and confirm that changes in temperature result in broad changes in activity in the brain.

A concern is how much is learned about the biology of the system. As the authors point out in the manuscript and associated citations, brain temperature is an important variable. It is already appreciated that temperature can increase as a function of brain activity and that fever and elevated brain temperatures can have significant pathological outcomes in human patients (see for instance 1,2).

- Much of their pronounced effects occur at extremely low temperatures (12-15 degrees Celsius). This is a temperature that would never be reached in physiological conditions (for rats or humans) and it is not surprising that at such low temperatures the authors see profound alterations to both neuronal and vascular activity. By contrast at more physiological ranges of temperature (see 5A for example) the measured variables such as tissue oxygenation are hardly different.
- Even in the case of their maximal responses (generally around 28 degrees), these temperatures are hardly physiological. The authors speculate in their discussion, "this may reflect an evolutionary conserved optimal temperature window that is ideal for sensory processing, and which permits associated dynamic changes in cortical temperature, as a result of metabolic processes and functional hyperaemia, to operate within physiological ranges". It is difficult to reconcile the notion that evolution has selected for an optimal temperature that is ten degrees below the actual temperature of the brain.
- Also the point of biological significance, some comment should be made about the relevance of choice of fit curves to their data. If the data is fit as well as they show by a simple relationship then some physiological basis for that specific relationship should exist.
- In the case of their measurements of evoked heating/cooling effects in 1E, the magnitude of heating and cooling around physiological temperatures (let's 37 +/- 4 degrees or so) is on the order of <.1 degrees. Also of question is what situations exist where the brain/core temperature differential is substantial enough to observe the magnitude heating/cooling effects they observe. For example, if there is generalized fever the brain will be at 40 degrees but so will the blood, thereby abrogating any cooling effects.

I would also caution against many instances in which the authors present correlational observations as potentially causal or overreach in their speculation. The following are some examples:

- The linkage between low frequency oscillation of the blood vessels observed in figure 3 and pathological outcomes. There is good evidence that elevated brain temperatures are more causally linked to pathology and given that low frequency oscillations in blood supply occur at normal temperatures why not just measure temperature directly? The observation that the low frequency band power increases at higher temperatures is just a correlative observation.
- The discussion of burst suppression and oscillation in blood supply is also strictly correlative. The claim that it contains a component that was “exacerbated by burst suppression activity” would require specific inhibition of burst suppression activity leading to suppression of this component.
- The claim that cortical interneurons are, “more adversely affected by temperature variations” and this helps explain reduced LFP/MUA activity at fever temperatures is also an over-reach. The authors make no measurement of interneuron activity – would they claim that the reduced activity measured at 12 degrees Celsius is also because of profound inhibitory neuron activity?

Minor points:

- For the above reasons, it may be relevant in the text of the manuscript to distinguish between “statistically significant” (which is frequently what the authors mean) and “biologically significant”. Typically the former is supported by the data in the manuscript while the latter has not been shown.
- The authors may consider citing Knutsen et al.³ in their discussion of the use of low temperatures to get more precise localization of neural activity. Presumably the mechanism at work here is reducing evoked neurovascular coupling leading to a localized increase in deoxyhemoglobin at the site of neural activity, which is exactly what Knutsen et al show.

1. Mrozek, S., Vardon, F. & Geeraerts, T. Brain Temperature: Physiology and Pathophysiology after Brain Injury. *Anesthesiol. Res. Pract.* 2012, 13 (2012).
2. Kiyatkin, E. A. Brain temperature and its role in physiology and pathophysiology: Lessons from 20 years of thermorecording. *Temp. Multidiscip. Biomed. J.* 6, 271 (2019).
3. Knutsen, P. M., Mateo, C. & Kleinfeld, D. Precision mapping of the vibrissa representation within murine primary somatosensory cortex. *Philos. Trans. R. Soc. B Biol. Sci.* 371, (2016).

We are grateful for the opportunity to respond to the insightful and positive comments made by both referees, and are delighted that these and the Editorial team found our manuscript to be of considerable interest. Below we provide point-by-point responses to the Editor's and reviewers' outstanding comments, and hope the manuscript now meets the Editorial threshold to be considered for publication.

Response to the Editor:

Please ensure that you tone-down conclusions throughout and very clearly discuss caveats and limitations of the study. We also request that you very clearly address the major concern about the biological relevance of the temperatures tested.

We thank the Editor for this recommendation. We have shortened our manuscript title to tone down the theragnostic implications of our work and also removed specific passages of text to the same aim (lines 345-348, see also lines 352-353, as requested also by Reviewer 2). We have also added additional text to discuss caveats of our analysis approaches (lines 323-325, as requested also by Reviewer 2) and substantially expanded the final Discussion paragraph (now subtitled 'Conclusions and methodological considerations, lines 359-386) which discusses the biological/translational relevance of our experiments and limitations of these (and see also amended text in lines 330-334).

At the same time, we ask that you ensure your manuscript complies with our editorial policies. Please see our revision file checklist for guidance on formatting the manuscript and complying with our policies. You will also find guidelines for replying to the referees' comments. You may also wish to review our formatting guidelines for final submissions here.

We have adjusted the format of the manuscript and believe it to comply with your editorial policies (e.g., Acknowledgments section, lines 621-633, now moved to after references, lines 834-846; Author Contributions section added, lines 848-854; Competing Interests section added lines 856-858).

Response to Reviewer #1

1. Did you try to mechanically stimulate the whisker? If you did, did you see any differences in the responses in comparison to whisker pad stimulation?

In this study we did not employ mechanical stimulation of individual whiskers as such a protocol would activate only an individual barrel region and yield a low signal to noise ratio, which would lead to floor and ceiling effects during the induced temperature changes. Rather, we employed stimulation of the whisker pad in order to induce a robust response in the entire barrel cortex and enable quantitative assessment of how graded brain temperature changes impacted functional responses.

2. Since temperature changes affect both spatial and temporal responses, have you tried to examine whether low and high temperature has an effect on the central-surround phenomena?

The challenging nature of our multimodal methodology unfortunately precluded us from feasibly placing an additional electrode in the surrounding regions (which is necessary for full

assessment of centre-surround neurovascular responses) and, as such, our study focuses on the primary neurovascular response in the stimulated region.

3. It's interesting that the Hbt onset time appeared to be different between 2s stimulation and 16s stimulation at low temperature. What are your thoughts on these differences?

Onset times were calculated using a standard approach by identifying 15% and 85% of the peak Hbt response and fitting a straight line between these points during the rising phase; where the linear function crossed the baseline was defined as the onset of the hemodynamic response. Hbt responses to longer duration stimulation appear to be more resilient at lower temperature in terms of time-to-peak kinetics than for shorter duration stimulation (take, for example, the ratio of time-to-peak to stimulus duration, see our Supplemental Table for the former), ostensibly due to the proportionally larger scale of vascular recruitment which is relatively more able to challenge the prevailing environmental conditions. As such, the linear function estimation of Hbt response onset for the 2s stimulation condition is less steep than that for the 16s condition, and is thus associated with a comparably earlier onset time. To improve clarity, we have now added text to the methods detailing the calculation of Hbt onset time (lines 565-569).

4. There is a "plateau" period after peak response seen with 16s stimulation and that period appeared to be diminished at a relative colder temperature (23-28°C). Would you care to discuss a bit about this?

The diminished plateau observed at ~23-28°C appears to arise as a result of these temperatures eliciting maximal responses, both in terms of hemodynamics and neural metrics, and the largest spatial hemodynamic response (see, for example, Hbt measures in Figure 3A). In the original manuscript, we speculated in the Discussion that this temperature window for maximal responses may be optimal for sensory processing and may allow dynamic changes in cortical temperature (due to metabolic processes and functional hypermia) to operate within physiological ranges. However, at the request of Reviewer 2, we have now removed this text in the manuscript (lines 345-348) as it extends beyond the interpretative scope of our experiments, although it is an intriguing observation which we will seek to elucidate in future work.

Response to Reviewer #2:

1. The authors convincingly demonstrate the ability to finely change cortical temperature (Fig 1C) and confirm that changes in temperature result in broad changes in activity in the brain. A concern is how much is learned about the biology of the system. As the authors point out in the manuscript and associated citations, brain temperature is an important variable. It is already appreciated that temperature can increase as a function of brain activity and that fever and elevated brain temperatures can have significant pathological outcomes in human patients (see for instance 1,2).

We thank the reviewer for their recognition of the methodological quality of our work and acknowledge that there is existing literature on the association between brain function and temperature and the negative impact of pathological brain temperatures. We believe that our present work goes beyond these aspects, however, in that we quantitatively and comprehensively describe the relationship between neurovascular responses and brain

temperature across a broad parameter space (i.e., temperature range). This, we feel, has enabled a critical, and to our knowledge novel, understanding of the effects of temperature modulation on brain function that is relevant for our understanding of neurovascular coupling, optimal methodological approaches, and the potential impact of brain temperatures that bidirectionally extend beyond normalcy during diagnostic procedures or in disease-associated states.

2. Much of their pronounced effects occur at extremely low temperatures (12-15 degrees Celsius). This is a temperature that would never be reached in physiological conditions (for rats or humans) and it is not surprising that at such low temperatures the authors see profound alterations to both neuronal and vascular activity. By contrast at more physiological ranges of temperature (see 5A for example) the measured variables such as tissue oxygenation are hardly different.

We recognise that the brain temperature modulation range in our study, by design, extends beyond physiological ranges. Nevertheless, there is a great deal of interest in the medical community with regard to focal brain cooling, not only as method by which to treat refractory epilepsy using implantable probes, but also to minimise ictal and interictal activity during intraoperative functional mapping using electrical stimulation, and which induce brain tissue temperatures at, and well below, 12-15°C (e.g. Ablah et al., 2009, *Seizure*; Bakken et al., 2003, *Journal of Neurosurgery*; Nomura et al., 2014, *Epilepsia*; Nomura et al., 2021, *Journal of Neurosurgery*). Moreover, there is growing interest in focal cooling, in of itself, as a novel and more precise method for intraoperative functional brain mapping and reversibly probing the relationship between neural dynamics and behaviour (Ibayashi et al., 2021 *World Surgery*; Long et al., 2016, *Neuron*; see also review by Bannerjee et al., 2021, *Neuron*), and in which tissue are cooled beyond physiological ranges. Thus, we feel that the cooler temperature inductions employed in our study, while not physiological in nature, are of direct relevance to these burgeoning clinical applications. We have now added additional text to the Discussion to emphasise the potential relevance of our findings (lines 367-377, and lines 294/295 and 302).

3. Even in the case of their maximal responses (generally around 28 degrees), these temperatures are hardly physiological. The authors speculate in their discussion, "this may reflect an evolutionary conserved optimal temperature window that is ideal for sensory processing, and which permits associated dynamic changes in cortical temperature, as a result of metabolic processes and functional hyperaemia, to operate within physiological ranges". It is difficult to reconcile the notion that evolution has selected for an optimal temperature that is ten degrees below the actual temperature of the brain.

We agree that the manuscript text highlighted by the reviewer is unclear and goes beyond the scope of the evidence provided; these speculative statements (lines 345-348) have therefore been removed for clarity.

4. Also the point of biological significance, some comment should be made about the relevance of choice of fit curves to their data. If the data is fit as well as they show by a simple relationship, then some physiological basis for that specific relationship should exist.

Curve fitting was performed by evaluating multiple functions and identifying that with the highest goodness-of-fit value. With respect to the notable inverted U-shaped relationships we observed, these suggest an intrinsic non-linear coupling between brain temperature and neurovascular responses, as well as the presence of an optimal brain temperature window for function (under our experimental conditions). At the reviewer's request, and for additional clarity, we have now added additional text to the Discussion on this aspect (lines 335-337).

5. Also of question is what situations exist where the brain/core temperature differential is substantial enough to observe the magnitude heating/cooling effects they observe. For example, if there is generalized fever the brain will be at 40 degrees but so will the blood, thereby abrogating any cooling effects.

We note that it has been reported that core temperature significantly underestimates brain temperature and fever in patients with acute neurological injury by up to several °C (Rumana et al., Crit Care Med; Rossi et al., 2001, *J Neurol Neurosurg Psychiatry*; Henker et al., 1998, *Neurology*; Schwab et al., 1997, *Neurology*; Mcilvoy 2004, *Journal of Neuroscience Nursing*). In addition, in preclinical rodent studies, localised brain heating of up to ~2°C can be induced by optogenetic stimulation (Owen et al., 2019, *Nature Neuroscience*) and near-infrared laser illumination during multiphoton microscopy (Podgorski and Ranganathan 2016, *Journal of Neurophysiology*), and during which core temperature is maintained normothermic. Thus, our findings are also likely to be of relevance for clinical situations involving pathological 'brain fever' and methodological approaches which induce regional brain/core temperature differentials. We have now added additional text to the Discussion on these implications for clarity (lines 333-334 and lines 367-375).

6. The linkage between low frequency oscillation of the blood vessels observed in figure 3 and pathological outcomes. There is good evidence that elevated brain temperatures are more causally linked to pathology and given that low frequency oscillations in blood supply occur at normal temperatures why not just measure temperature directly? The observation that the low frequency band power increases at higher temperatures is just a correlative observation.

As described in Response #5 above, core temperature may not act as a faithful proxy of brain temperature in several neurological conditions, which may lead to neural hyperthermia being undetected and potential worsening of neurological injury. Since intracranial implantation of temperature probes to provide an unequivocal readout of brain temperature may not be a practicable approach in many situations, we suggest that that the observed low-frequency oscillation may be a promising non-invasive functional biomarker of pathologically elevated brain temperature. We most certainly agree with the Reviewer's general view pertaining to the pitfall of confounding correlation with causality, however, in this instance, we feel that our finding goes beyond a basic correlative observation, since we experimentally manipulated brain temperature bidirectionally which revealed the emergence of the low-frequency oscillation at elevated brain temperatures, and which differed spectrally from 'physiological vasomotion' (centre frequency 0.1Hz).

7. The discussion of burst suppression and oscillation in blood supply is also strictly correlative. The claim that it contains a component that was "exacerbated by burst suppression activity" would require specific inhibition of burst suppression activity leading to suppression of this component.

We agree with the reviewer that this observation is correlative; our reporting of this was motivated by our intention to provide underlying insights into this observation. In light of the reviewer's point, we have substantially de-emphasised this point in the discussion (lines 323-325) and clearly stated this to be correlative analysis in the results (line 226).

8. The claim that cortical interneurons are, "more adversely affected by temperature variations" and this helps explain reduced LFP/MUA activity at fever temperatures is also an over-reach. The authors make no measurement of interneuron activity – would they claim that the reduced activity measured at 12 degrees Celsius is also because of profound inhibitory neuron activity?

Our citing of the Motamedi et al. articles was intended to provide context and discuss the intriguing reports that interneurons may be more susceptible to brain temperature changes. However, we fully take the Reviewer's point that these alone do not fully explain our observations and have thus amended the text for clarity (line 352).

9. For the above reasons, it may be relevant in the text of the manuscript to distinguish between "statistically significant" (which is frequently what the authors mean) and "biologically significant". Typically the former is supported by the data in the manuscript while the latter has not been shown.

We thank the reviewer for making this important point and have endeavoured to increase the clarity in the revised manuscript by stating "statically significant" (e.g., lines 118 and 150).

9. The authors may consider citing Knutsen et al. in their discussion of the use of low temperatures to get more precise localization of neural activity. Presumably the mechanism at work here is reducing evoked neurovascular coupling leading to a localized increase in deoxyhemoglobin at the site of neural activity, which is exactly what Knutsen et al show.

We thank the reviewer for highlighting this article and which has now been included in the revised manuscript (line 284).

REVIEWERS' COMMENTS:

Reviewer #1 (Remarks to the Author):

The authors have addressed all my comments. The manuscript is suitable for publication.